# Patient and Family-Centered Care to Promote Inpatient Safety: An Exploration of Nursing Care and Management Processes

**DOI:** 10.3390/nursrep15070260

**Published:** 2025-07-16

**Authors:** Tânia Correia, Maria Manuela Martins, Fernando Barroso, Lara Pinho, João Longo, Olga Valentim

**Affiliations:** 1Institute of Biomedical Sciences Abel Salazar of the University of Porto (ICBAS), Oporto University (UP), 4050-313 Porto, Portugal; 2RISE-Health, Nursing School of Porto (ESEP), 4200-450 Porto, Portugal; ovalentim@esel.pt; 3School of Health Sciences, Polytechnic Institute of Viseu, 3500-843 Viseu, Portugal; 4Nursing Department, Nursing School of Porto (ESEP), 4050-313 Porto, Portugal; 5Setúbal Hospital Center, 2910-446 Setúbal, Portugal; 6Nursing Department, Évora University, 7000-811 Évora, Portugal; 7Comprehensive Health Research Centre (CHRC), Évora University, 7000-811 Évora, Portugal; 8Nursing Research, Innovation and Development Centre of Lisbon (CIDNUR), 1600-096 Lisbon, Portugal; 9Department of Mental Health and Psychiatric Nursing, Nursing School of Lisbon (ESEL), 1600-096 Lisbon, Portugal

**Keywords:** family nursing, family-centered care, hospitalization, patient safety, safety management

## Abstract

**Background**: Family involvement in promoting patient safety is a strategy that is increasingly recognized. Nurses have a major role to play here. This study aims to know the family care process developed by nurses and explore the logistic process identified as support for the development of family care in ensuring patient safety in the hospital. **Methods**: An interpretative qualitative study was conducted through semi-structured interviews with 10 nurses selected by convenience. Content analysis was performed using Atlas.ti 9.1.7 software and Bardin’s methodology. **Results**: Two large *families* were identified according to the nature of the strategies mentioned by the participant/s: *assistance process and logistic process*, 5 categories and 23 subcategories. *Care process* categories: *initial assessment*, *planning*, *and implementation*. Categories of the logistic process: *material and human resources and organization*. **Conclusions**: To implement patient and family-centered care to ensure patient safety, it is necessary to adjust the care and logistic process. At the care level, the importance of the nursing process in the organization of care is evident. At the logistical level, organizational initiatives are needed to stimulate and support this philosophy of care and to intervene at the level of human and material resources.

## 1. Introduction

Hospitalization is always a time of anxiety, stress, insecurity, and anguish for both the patient and their family [1,2]. The family can play a key role in contributing to the patient’s emotional stability and providing support across various domains [3,4]. This involvement also has a positive impact on the family’s own well-being, as it fulfils their need to support the patient, access information directly, and be present with their hospitalized relatives [3,4,5]. Therefore, it is essential to implement models of care that take into account the needs and rights of both patients and their families [3].

In alignment with the humanization of hospital health care, the patient and family-centered care (PFCC) model has been recommended as a best practice for including the family in care provision [4,6]. This model represents an approach to the planning, implementation, and evaluation of care that brings benefits to both patients and families. It recognizes the family as an integral component of care and as a partner, particularly in decision-making, through information sharing, respect, dignity, negotiation, participation, collaboration, and a shift in professionals’ attitudes towards patients and their families [6,7]. This partnership with families, at clinical, strategic, and policy levels, has been supported by evidence showing that it is essential for ensuring the quality and safety of health care [6,8]. The Global Patient Safety Action Plan 2021–2030 recognizes that the involvement of patients and families in the care process is essential to achieving safer care. It offers concrete recommendations, including involving patients and families in the formulation of patient safety policies and ensuring the transparent sharing of information [8].

A meta-analysis of 22 studies on patient and family engagement interventions demonstrated a significant reduction in adverse events, a decrease in length of hospital stay, improvements in patients’ safety experience, and increased satisfaction with care [9].

The World Health Organization defines safety as “the reduction of the risk of unnecessary harm to an acceptable minimum. An acceptable minimum refers to the collective notions of given current knowledge, available resources, and the context in which care was provided, weighed against the risk of non-treatment or other treatment” [10].

The Patient Safety Rights Charter affirms the right of patients and their families to participate in ensuring care safety [11]. Family members designated by the patient may actively engage in decision-making and in identifying potential risks. They can contribute to the improvement of the health system by participating in awareness campaigns, policy development, monitoring, evaluation, and research, including as members of advisory committees or health councils [11].

Despite its acknowledged benefits, the PFCC model is not yet widely implemented in health services, resulting in a gap between current practice and recommended standards [12]. Nurses hold ambivalent views regarding the role of the family in the care of hospitalized patients. On the one hand, they recognize the family’s importance in the care process; on the other, they perceive the family as an additional source of workload and time investment, and as a potential risk to patient safety [12,13,14].

Available evidence indicates that when health managers, professionals, patients, and families work in a partnership, the quality and safety of care improve, satisfaction among patients, families, and professionals increases, and health care costs are reduced [6].

Published studies also show that families often act to protect the safety of their hospitalized relatives and, in some cases, succeed in following patient safety recommendations [12,15,16]. However, families frequently report feeling unprepared and lacking in guidance and collaboration from health professionals [12,17].

The available evidence highlights the need for nurses and other health professionals to regard the family as a partner in care, rather than as a supervisory or disruptive presence. Health professionals should adopt attitudes that promote and facilitate family involvement in the care process. To achieve this, it is essential to strengthen communication between health professionals and families concerning patient safety during hospitalization [12].

The available evidence evaluates patient and family involvement only at the level of direct health care delivery [18]. However, there are few published studies on the involvement of patients and families in patient safety [14]. The application of the PFCC model in adult care requires further research and implementation when compared to its use in pediatric settings. Given that nurses are the health professionals who maintain the closest and most continuous contact with families, their behaviors and attitudes can significantly influence the hospitalization experience of both patients and their families, as well as key outcomes related to care quality and safety. Recent evidence underscores that nurses, as the primary formal caregivers, should play a leading role in promoting patient and family involvement in safety-related practices [18].

Research on patient and family involvement in patient safety remains limited, particularly at the organizational and system levels [18,19]. Therefore, the aim of this study is to understand the process of family care in the hospital setting as developed by nurses, and to explore the logistical processes that support the implementation of family-centered care to ensure patient safety during hospitalization.

## 2. Materials and Methods

This qualitative, interpretative study aimed to understand the process of family care in the hospital setting as developed by nurses and to explore the logistical processes identified as supporting the development of such care to ensure patient safety.

Participants were nurses working in internal medicine and surgical wards across four hospitals in northern Portugal. They were selected through convenience sampling, as they were the most accessible and met the pre-established inclusion criteria. These criteria included the following: being in active clinical practice at the time of the interview; having experience in internal medicine and/or surgery; having at least four years of hospital-based professional experience; and being available to participate. After the seventh interview, no new data emerged that contributed to the development of additional categories. Therefore, after completing ten interviews, data saturation was considered to have been reached [20,21].

Based on the criterion of theoretical saturation, a total of 10 participants were included: 9 female nurses and 1 male nurse, aged between 28 and 62 years (mean age 39 years). The age distribution was as follows: one participant under 30 years, six between 30 and 40 years, two between 40 and 50 years, and one over 60 years. Their professional experience ranged from 6.5 to 39 years, with a mean age of 16 years. Seven participants held master’s degrees, and nine were specialists—six in rehabilitation nursing, two in medical-surgical nursing, and one in mental health and psychiatric nursing (Table 1).

Data were collected through individual semi-structured interviews conducted between June and September 2020. Participants frequently responded to questions even before they were formally asked, which confirmed that the semi-structured interview format was the most appropriate method for this study.

Data were collected through individual semi-structured interviews between June and September 2020. Participants often answered the questions even before they were asked. Therefore, the semi-structured interview was the best option.

The interview script (Appendix A) resulted from an integrative review conducted a phase prior to this study [12]. This guide was pilot tested, and no modifications were necessary. Therefore, it was used for all interviews by the same researcher to ensure consistency and reliability in data collection.

Each interview was recorded and transcribed in full, and its transcription was sent to the interviewee for validation.

Content analysis was conducted following Bardin’s framework, which comprises three phases: pre-analysis, exploration, and finally, the treatment of results and interpretation [22]. The data analysis was carried out by T.C. and M.M., and the reflection on the findings was further expanded through discussions with F.B. and O.V.

During the exploratory phase, common content was identified, allowing for the emergence of thematic areas, designated as categories. The Atlas.ti^®^ 9.1.7 software was used for coding and categorizing the interview data. This software facilitated the creation of units of analysis (categories) and their identification from significant units extracted from the interviews, which were subsequently grouped into broader thematic families. This process enabled the generation of the study’s results. We declare that this report adheres to the COREQ guidelines for qualitative research reporting [23].

All ethical and legal principles governing research involving human participants were rigorously respected throughout the study. Ethical approval was obtained from the Joint Ethics Committee of the Porto Hospital and University Centre and the Abel Salazar Biomedical Sciences Institute (ICBAS) at the University of Porto (UP)—protocol code 2020/CE/P010/P320/CETI/ICBAS. The study procedures strictly upheld the principles of participant anonymity, confidentiality, and voluntary informed consent. Furthermore, the research was conducted in full compliance with the ethical standards outlined in the Declaration of Helsinki. Participants received detailed information about the study’s objectives, procedures, potential risks and benefits, confidentiality measures, and their right to withdraw at any time without penalty.

## 3. Results

Based on the analysis of the data obtained from the nurses’ interviews, five categories (codes) were identified and grouped into two major thematic areas (families) according to the nature of the strategies described by the participants: care process and logistic process (Table 2 and Table 3). Within each category, a total of 23 subcategories were identified.

In the care process family, three categories emerged: initial assessment, planning, and implementation. These categories are part of the nursing process, a systematic approach grounded in scientific methodology that contributes positively to the quality and safety of nursing care. The nursing process consists of five stages: initial assessment, diagnosis, planning, implementation, and final evaluation [24].

The analysis of the interview data revealed two main thematic families: the **care process** and the **logistic process**, which emerged as the two major strategic domains for implementing patient and family-centered care (PFCC) with a focus on patient safety (Table 2).

The **care process** encompasses a series of interventions related to patient diagnosis and treatment, aimed at providing optimal care [25]. Studies have shown that improving the organization of care delivery contributes to enhanced patient safety in healthcare settings [25]. Furthermore, there is a strong correlation between the quality of the care process and the centrality of the patient and family in care [25,26].

In contrast, the **logistic process** in healthcare refers to the coordination of complex operations involving multiple people, facilities, and resources within the health system [27]. It also plays a crucial role in supporting the quality of the care process [27].

As illustrated in the Figure 1, the **care process** is the most visible component when it comes to the practical application of PFCC in a safe manner, as it involves direct interventions with patients and families. The **logistic process**, although less visible by nature, is essential to the structure and organization of healthcare and, in this context, to enabling and sustaining safe PFCC.

## 4. Discussion

### 4.1. Care Process

In the **care process** family, three categories were identified in alignment with the stages of the nursing process: **initial assessment**, **planning**, and **implementation** [24].

The **initial assessment** involves collecting information about the person, including physical and psychosocial data. This may be obtained through interviews, observation, physical examinations, review of clinical records, and conversations with family members. It is a continuous and structured process aimed at understanding the person and their needs. This information is essential for establishing nursing diagnoses and planning care accordingly [28]. In this category, the initial assessment, three subcategories were identified: assessment of the family/identification of the relative of reference, identification with the patient of the people authorized to visit and schedules, and collecting information about the patient from the family.

Participants highlighted the importance of **assessing the family/identification of the relative of reference** at the beginning of care, as the level of family involvement can vary significantly. Not all families are available, emotionally close, or willing to take part in the care process, and this decision must be respected. Therefore, it is essential to evaluate, on a case-by-case basis, whether the family is willing to participate and whether the patient agrees with their involvement. When there is mutual agreement, this partnership can be highly beneficial for all parties involved. Therefore, assessing each family is considered a fundamental first step in establishing a safe and beneficial care partnership, grounded in respect for their autonomy [29]. The implementation of this assessment to evaluate the family’s willingness and capacity to participate in PFCC is not clearly evident in practice. Therefore, we consider this a significant finding of the present study.

Furthermore, participants emphasized the importance of identifying the reference relative, chosen by the patient, regardless of whether they are a direct family member or a significant person from the patient’s support network. A family caregiver is defined as a relative or friend who voluntarily accompanies the hospitalized patient to provide physical, psychosocial, or spiritual support [30]. Although the identification of such a caregiver is recommended, there is a lack of evidence in the scientific literature supporting its effectiveness in enhancing the safety aspect of patient and family-centered care [31].

Additionally, within the scope of the initial assessment, participants highlighted the importance of **identifying**, **together with the patient**, **the individuals authorized to visit** and defining appropriate visiting schedules. Participants suggested that, ideally, upon admission, patients should have the opportunity to indicate who they wish to receive as visitors, the number of visits, preferred times, and the type of care they would like their visitors to assist with. The identification of visitors by patients themselves is clearly an important contribution to safety, as it helps prevent exposure to individuals who may be irrelevant or potentially harmful to their well-being. Additionally, knowing when patients prefer to receive visitors supports more effective care planning. Although participants recognized the value of this measure, it does not appear to be widely implemented, despite being a long-standing recommendation for enhancing the quality and safety of care [31].

**Collecting information about the patient from the family** was mentioned as essential and even an important difficulty during the period of restricted visits due to COVID-19. The participants mentioned that family members serve as an important link between the healthcare team and the patient, facilitating the collection of essential information about the patient’s medical history and daily habits, information that is crucial for patient safety, and that this was especially challenging during the COVID-19 period due to the absence of family presence [5,17]. As proof of the importance of the family’s information and knowledge of the patient, a study conducted in 2017 developed a family reporting methodology that suggests that families can be important partners in patient safety through their prior knowledge of the patient and serve as unique hospital accompaniment [32].

Care **planning** is based on previously collected information and involves defining nursing interventions to be adopted, identifying expected results and establishing priorities [28]. In this category, planning, it was possible to distinguish four subcategories: discussing with the family the care to be developed in partnership, and preparing discharge with the family from the beginning of hospitalization.

A subcategory emerged regarding **discussing with the family the care to be developed in partnership**, although it was mentioned infrequently. Participants highlighted that when both the patient and family agree, this collaboration is beneficial for everyone. They also gave examples of respecting patient preferences, such as choosing the timing of personal care. This family-centered planning approach, which prioritizes the preferences and capabilities of the family, aligns with the recommendations of the Institute for Patient and Family-Centered Care. It encourages hospital services to support and involve family caregivers in presence, planning, decision-making, and care based on the preferences expressed by patients and their families [31]. Research findings indicate that nurses often have unclear or ambiguous views regarding the role that family members should assume in the hospital setting. This uncertainty leads to insufficient communication with families, resulting in a lack of understanding about how they should act, the patient’s needs, and ways to contribute positively to care. Consequently, this gap can cause issues in patient care, including safety risks and conflicts with nursing staff. Therefore, this research highlights the necessity for clearer and more effective negotiation of family members’ roles within the care process [5,18,19,33].

The subcategory **prepare discharge with the family from the beginning of hospitalization** was strongly emphasized by participants. They highlighted the importance of involving and empowering families early on to build trust and ease difficulties. Nurses use moments of care to engage families, encouraging their participation in activities such as mobility support, which helps families feel useful and contributes to patient care. These small actions collectively prepare both the patient and family for a smoother discharge process. Current recommendations state that discharge planning should begin within 24 h of hospital admission, be well-structured, identify problems and opportunities, involve all relevant parties, and be communicated clearly to both the patient and their family [34,35].

The participants also linked discharge preparation involving the family to patient safety during the transition home. They emphasized the importance of educating families about necessary adaptations in the home environment, such as reorganizing spaces to reduce fall risks, especially for patients with mobility challenges. This proactive approach aims to ensure a safer discharge and support ongoing recovery. This is one of the strategic elements for the intervention centered on the family with the objective of patient safety in the period after discharge [36]. According to the available evidence, discharge planning is essential to ensure patient safety. When there are failures at this level, adverse events may arise post-discharge. The most known and evidenced adverse events are medication errors [37]. In this context, the family has an important role. Nurses should consider the family in the preparation for discharge in order to maintain continuity and safety of care when the person returns home [38].

**Implementation** refers to the effective delivery of care and comprises the direct execution of care, teaching, guidance, and/or referral [28]. This category, **implementation**, includes the following subcategories: teaching/informing the family, providing family participation in care, supervision/accompaniment of the family in the hospital by nurses, encouraging the family to report safety issues, and making video calls.

The participants emphasized that **teaching/informing the family** is essential to ensure that their presence is safe for the hospitalized patient. They also highlighted it as a strategy to promote family involvement, guiding them so that their participation contributes positively to safety during care delivery. The relationship between the failure of professionals to communicate with the patient and family and patient safety emerges with little evidence. Giving information to patients and families about the health process, diagnosis, treatment, and prognosis is uncomfortable for some health professionals. They only share information that they consider necessary, hiding behind the indications of institutional data protection [33,39,40]. Studies with interventions based on family communication and education resulted in improvements in patient literacy and safety outcomes [6,12].

The subcategory **providing family participation in care** was mentioned as important not only with children, but also with adults. Participants noted that while family participation is often encouraged in pediatric care, it is less common with adults and the elderly. They emphasized the need to extend this practice to all patients, as family members can provide valuable insights into the patient’s preferences and contribute positively to care, especially in preparation for discharge. Family caregivers play an important role in caring for the elderly or chronically ill person at home. Thus, in order to maintain the continuity of care, there is a clear need to involve the family caregiver as a partner in hospital care [38,41].

**Supervision/accompaniment of the family in hospital by nurses** emerged, in which the participants mentioned that that nurses should be responsible for supervising and accompanying family members during hospital visits, as they are the professionals best equipped for this role. They described how this supervision involves welcoming relatives, providing guidance on infection prevention measures, such as hand hygiene and limiting contact with hospital equipment, and offering practical advice about personal belongings. This supervision aims to ensure that the family’s presence is safe for the patient and aligned with hospital protocols. This strategy appears to be particularly useful for families who are still acquiring knowledge about healthcare and require additional guidance. As this family-centered care approach is not yet well-documented in the existing literature, it stands out as a potentially innovative finding that warrants further investigation in future studies.

The subcategory **encouraging the family to report safety issues** was highlighted by some participants as a relevant strategy to promote patient safety. They recognized that family members and patients can play an important role in identifying potentially unsafe situations or errors, acting as an additional safety check within care processes. This proactive involvement contributes to the early detection of problems and supports safer clinical practices. The available evidence shows that relatives report undetected/occult events, avoidable adverse events not detected otherwise or by other elements of healthcare. Thus, they represent a unique and very useful source of safety information [32].

**Making video calls** was identified by the participants as a strategy used to maintain the connection between the patient and their family during hospitalization. They referred to the possibility of making video calls at the patient’s or family’s request, usually on a daily basis or whenever necessary, as a way to promote emotional closeness and support during the hospital stay. The use of digital technology for involving the family in hospital health care, although not statistically significant, has shown a relationship with increased satisfaction of the patient and family, decreased anxiety, and improved family involvement [42].

From the analysis of these results, we highlight that no implementation strategies for the PCF were identified in all steps of the nursing process; namely, in the scope of diagnosis and final assessment. This finding is consistent with other studies on the application of the nursing process in Portuguese hospitals, which reveal that, despite recognition of its value, its implementation is limited, partial, and superficial. Nurses tend to adopt an approach focused on controlling signs and symptoms, guided by routines and traditions rather than systematized care. Bureaucratic tasks and various complementary functions occupy nurses, making it difficult to properly implement the systematization of nursing care (SNC) [43,44]. Within the framework of family-centered care, as in all care approaches, it is essential that the nursing process be fully applied, following all its phases. However, no available evidence was found regarding the comprehensive implementation of the nursing process in the context of family-centered care.

### 4.2. Logistic Process

The **logistic process**
*family* includes the categories ***human and material resources*** and ***organization***. To ensure the provision of healthcare, hospitals require two essential resources: human and material. In order to guarantee the quality of care, it is crucial that these resources are used efficiently and effectively [45]. In turn, health workers are the lifeblood of health systems. This requires investment, planning, education, recruitment, management, and stimulation of human resources for health to meet present and future health challenges [46].

In the **human and material resources category**, the following subcategories were identified: developing leadership influencing PFCC, promoting safety culture in the hospital, planning training for nurses on the attitude of involvement/communication attitude, improving nursing staffing, and creating spaces outside the ward for visits.

The participants identified the **development of leadership to influence** PFCC as a necessary strategy that is currently lacking. The scientific evidence identifies leadership as a key strategy for the implementation of the PFCC [47,48]. Even so, the relationship between an developing leadership influencing PFCC and patient safety is not evident in the scientific literature and therefore needs further research investment.

**Promoting safety culture in the hospital** is identified as a contribution to the safe CFP as a strategy to influence and set an example for the family that attends the hospital. Research results show that having patients and families as partners and agents promoting safe care is essential for a safety culture [39].

Participants emphasized the need for **planning training for nurses on the attitude of involvement/communication attitude** and fostering a more engaging and available attitude toward families, noting that traditional training methods have been largely ineffective. As previously mentioned, the lack of communication with the family can lead to problems for patient safety [12,49]. Training in this area, besides contributing to the effective implementation of the PFCC, contributes in the same measure to patient safety [4].

**Improving nursing staffing** was referred to as crucial to safely implementing the PFCC by participants. Low nursing ratios are related to higher patient mortality [50]. Similarly, low nursing ratios represent a barrier to implementing PFCC in a safe manner [48].

**Creating spaces outside the ward for visits** is the subcategory mentioned as a reinforcement for the safety of the family’s presence in the hospital with the patient. The need for physical and architectural structures that facilitate family-centered care is clear and well recognized, especially by health managers. However, from a provider-centered perspective, there is often a noted lack of available space for private conversations with patients and their families [48]. This highlights the importance of creating dedicated family spaces that prioritize the family’s needs and privacy, even within the hospital environment.

Healthcare **organization** is closely related to healthcare management and concerns the specific processes of the healthcare system and involves standards of good practice, as well as stimulation of initiatives to encourage high quality and effectiveness of healthcare [51]. This last category encompasses the following subcategories: stimulating the primary nursing method, stimulating family care centered on the family member of reference, providing lockers for the family, creating a checklist for teaching the family, create a guide for the family in hospital, visit management, separate visiting hours for family members of reference, extend visiting hours, and a 24 h visiting policy.

**Stimulating the primary nursing method** is recognized by participants as a nursing care organization methodology that facilitates a more effective and safer PCF. In this context, streamlining the primary nursing method is clearly recognized as a strategy offering multiple benefits, including enhanced communication between the healthcare team and the family, as well as better discharge preparation. From the perspective of patient safety, these aspects are particularly critical [52].

Similarly, the subcategory of **encouraging family-centered care focused on the primary family caregiver** also emerges. The participants mentioned that they encourage families to appoint a spokesperson, directing communication primarily to the family member designated by the patient to ensure clear and organized information sharing. Beyond identifying the designated family member with the patient, this category involves focusing communication primarily on that relative. While balancing communication with the entire family and prioritizing the family caregiver, the latter appears to offer greater safety. However, there is currently no evidence in the scientific literature supporting this approach as beneficial for safe family-centered care, highlighting the need for further research in this area.

The subcategory of **providing lockers for the family** as a strategy to accommodate the family in the hospital to prevent them from leaving their belongings around the nursing unit and safeguarding hospital infection control. This strategy is also not evidenced in the scientific literature. Considering its apparent relevance, further studies are needed to identify the benefits of this strategy for patient and family safety in the hospital.

**Creating a checklist for teaching the family** is indicated as an organizational strategy to empower the family for patient safety. Still, it has shown good results. It allows for systematizing routine procedures, improving communication between health professionals, and has proven to be effective in reducing errors, complications, mortality, and hospitalization time [53]. At this level, scientific literature already identifies benefits of patient safety checklists with positive impact on patient empowerment and engagement in safety-related behaviors [53]. Given the potential for family involvement in these strategies, the involvement of the family in completing safety checklists may represent an added value to be studied.

**A guide for the family in the hospital** is mentioned by the participants as a structure for the integration of the family in this environment. With this strategy, we created another means to increase literacy about health care in the hospital, namely about health safety [54].

In the subcategory **visit management**, participants expressed ambiguity, with some stating that visits should be managed by the medical team, the nursing team, or a combination of both. There is no evidence in the literature about which health professional is best suited to perform this management. However, management implies monitoring, coordination, administration, planning, and organization of activities and resources to achieve specific goals [55]. In this case, the purpose would be to ensure the patient’s safety during hospital visits. In this context, nurses are professionals who are continuously present with the patient. Visit management software already exists where it is possible to define who is authorized to visit the patient. In line with the results already mentioned in this study, this list should be made with the patient and family under the guidance and supervision of a health professional, who may be the nurse. The debate regarding whether nurses or physicians should manage hospital visiting policies remains unresolved. It is suggested that clearer policy recommendations could help to clarify this issue.

Regarding visiting hours, three distinct but potentially complementary subcategories emerged: **separate visiting hours for family members of reference**, **extended visiting hours**, **and a 24 h visiting policy**. Studies have shown that 24 h open visitation can improve outcomes and increase patient satisfaction, with little or no increase in security incidents and minimal disruption to facility operations [56]. As previously mentioned, evidence shows that an informed family, with health safety literacy, may even become an important contributor to patient safety, in this case 24 h/day.

From the results of this study, the number of subcategories associated with the logistical process was higher than the number of subcategories identified in the care process. Thus, the participating nurses consider that there is a greater need for structural and logistical support for the implementation of the PFCC, and not only to expect that the intervention of the nurses who provide care happens according to the philosophy of the PFCC in a safe way. This is something new in view of the available evidence with strategies that do not appear with evidence in the scientific literature. The known strategic recommendations for the implementation of the PFCC focus mainly on care delivery, with limited guidance on institutional and management support measures [57]. This demonstrates that health managers have a key role to play in the implementation of the safe PFCC.

This is a qualitative study that allowed identifying several strategies for the implementation of the PFCC for patient safety in both the care process and the logistics process. Thus, these interventions and strategies have the potential to contribute to the PFCC for patient safety and should be considered by both care nurses and nurse managers. Further studies on each of them and in different realities are needed to validate their contribution to the implementation of the PFCC for patient safety.

### 4.3. Implications for Clinical Practice

For the implementation of the PFCC to ensure patient safety, adjustments and initiatives are necessary at the care process level, but also at the level of the logistic process.

At the care level, the importance of the nursing process oriented towards the PFCC with patient safety is highlighted. This should be the basis for integrating the PFCC, ensuring systematization and rigor for safer care. Significant examples are the need to discuss with the family the care to be developed in partnership, prepare discharge with the family from the beginning of hospitalization, teach/inform the family, provide family participation in care, and encourage the family to report safety issues, among other interventions presented in this research work.

At the logistical level, organizational initiatives are needed to stimulate and support the BCPS for patient safety, as well as to intervene at the level of human and material resources in the same direction. Examples of this are the need to develop leadership that influences PCF, promote a safety culture in the hospital, plan nurses’ training on the attitude of involvement/communication with the family, improve nursing appropriations, and create spaces outside the ward for visits, among other interventions identified in this article. Therefore, nurse managers also have an important role that is valued by the practice nurses in this study.

The assistance process is the most evident of the organic of the PFCC with security. It concerns the direct intervention with the person and the family. In turn, the logistic process is not so visible, but it is fundamental to the structure and organization of health care and, in this case, to encourage and structure the safe PFCC.

## 5. Conclusions

The participants’ accounts allowed us to understand the complexity of the care process when family members are present and its implications for nurses’ work to ensure patient safety. The importance, and the need, for an active role of care nurses and nurse managers in the implementation of the PFCC with a view toward patient safety, is clear.

The care process is guided by an organization where assessment, planning, and intervention are valued, while diagnosis and assessment of nursing care results are not represented.

The results of this study show a greater need for strategies by the logistic structure for the operationalization of patient and family-centered care for patient safety. This is an important finding since the recommendations in this area are more focused on care delivery, placing more responsibility on nurses and other direct care health professionals.

The analysis of the speeches makes us realize that the health manager may play a key role, particularly in the organization and leadership of the logistic process for a patient and family-centered care for patient safety.

The findings suggest that policies should formalize family involvement in care, promote structured communication, and ensure infrastructure that supports privacy and convenience. Training should focus on developing nurses’ communication and leadership skills, integrating family-centered care into all phases of the nursing process, and preparing families for safe post-discharge care. These measures can strengthen patient safety and the quality of care.

The limitation of this study is that data collection was carried out during a pandemic period with restricted hospital visits, which may have influenced the participants in their answers. On the other hand, the families’ absence may have been perceived by the nurses as negative for their practice and this influence may have been positive for these results. Due to the small sample size, the transferability of the findings is limited. However, the depth of the participants’ experiences offers valuable insights that may be applicable to similar hospital settings or inform future research and policy development in family-centered care. Thus, we suggest the replication of this study in another period and with participants from different institutional realities.

The availability of data implies the identification of hospitals, services, and nurses participating in the study. In respect of confidentiality, data are not made available beyond what is presented in the study. This may be considered another limitation of the study.

Further research should focus on the effects of collaboration between family caregivers and nurses as part of care with the aim of promoting patient safety, namely focusing on each of the strategies identified in this study in different realities and thus validate their contribution to patient and family-centered care for patient safety. Specifically, the strategies for which there is no available evidence supporting their effectiveness, such as providing lockers for family members and encouraging family-centered care focused on the primary family caregiver, emerged as innovative findings of this study. Further research is needed to explore their impact on patient safety.

## Figures and Tables

**Figure 1 nursrep-15-00260-f001:**
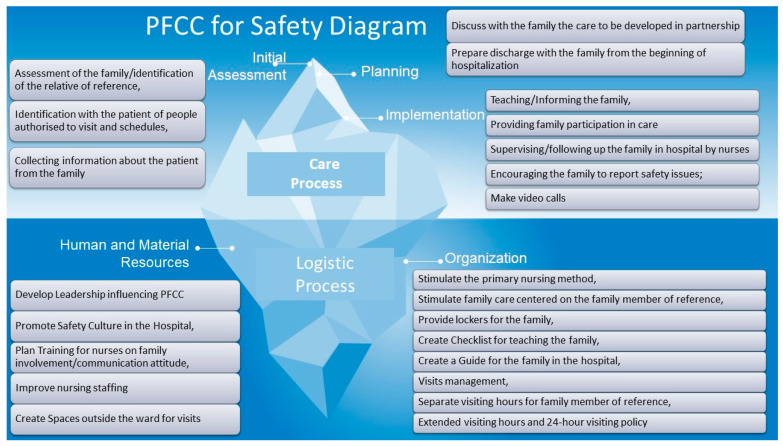
PFCC for Safety Diagram.

**Table 1 nursrep-15-00260-t001:** Participant Data.

Participants	
*n*	10
Gender	9 females
	1 male
Age	between 28 and 62 years (average 39 years)
Specialists	9
Masters	7

**Table 2 nursrep-15-00260-t002:** Results of data analysis: families, categories, and subcategories.

Families	Categories	Subcategories	*n*	Analysis Weight
Care process	Initial assessment	Assessment of the family/identification of the relative of reference;	22	22.30%
		Identification with the patient of people authorized to visit and schedules;	5	
		Collecting information about the patient from the family.	2	
	Planning	Discuss with the family the care to be developed in partnership;	2	5.4%
		Prepare discharge with the family from the beginning of hospitalization.	5	
	Implementation	Teaching/Informing the family;	13	25.38%
		Providing family participation in care;	5	
		Supervising/following up the family in hospital by nurses;	5	
		Encouraging the family to report safety issues;	4	
		Make video calls.	6	
Logistic process	Human and material resources	Develop leadership influencing PFCC;	2	17.69%
		Promote safety culture in the hospital;	2	
		Plan Training for nurses on family involvement/communication attitude;	3	
		Improve nursing staffing;	6	
		Create spaces outside the ward for visits.	10	
	Organization	Stimulate the primary nursing method;	2	32.30%
		Stimulate family care centered on the family member of reference;	11	
		Provide lockers for the family;	2	
		Create checklist for teaching the family;	2	
		Create a guide for the family in the hospital;	1	
		Visits management;	15	
		Separate visiting hours for family member of reference;	2	
		Extended visiting hours and 24 h visiting policy.	3	

**Table 3 nursrep-15-00260-t003:** Results of data analysis: subcategories and quotations.

Subcategories	Quotations
Assessment of the family/identification of the relative of reference	*“… we won’t have the same level of involvement in all families. There are children who don’t want to take care of their parents because they have never been close, for example. We cannot expect that all families will collaborate in the provision of care, we are there for that. This evaluation should be done by us on a case-by-case basis. If the family is willing, if the patient, being conscious and oriented, also agrees, it is a very advantageous partnership for all. And in this whole context, each family has to find its own balance. Each family is a family” (E3)* *“…There are those elderly people who had never been bed-ridden, were independent, and the daughter is even available, but does not feel at ease for this proximity. There are also ulcer treatments, there are families who even want to see and there is no problem if they want to see. On the other hand, there are other families who feel uncomfortable with the smells or with the size of the wound. It is necessary to adapt to each situation. But I think above all we have to ask the patient and the family if they want to stay to watch or participate.” (E4)* *“… regardless of whether it is a relative or a significant person as long as it is the patient who nominates or is the main caregiver… The patient may want someone who is not a direct relative but who they consider family, someone closer, a friend.” (E4)*
Identification with the patient of people authorized to visit and schedules	*“The ideal would be when the patient is admitted, when he/she enters we could even create a list of people that he/she would like to be visited… We have to create a structure that allows the patient to give his/her opinion and say by whom he/she wants to be visited, the number of visits, the schedule, and in what care he/she would like them to help”. (E3)*
Collecting information about the patient from the family	*“It is part of our initial assessment to ask some questions related to safety, such as the risk of falling, the risk of escape in patients with psychiatric disorders, the risk of pressure ulcers in bedridden patients, allergies. Closed questions, but it could be possible to ask questions in this area.” (E10) * *“… in the first approach to the patient and family we talk about the risks that the patient runs, the information that the patient and family provide is precious, in the strategies to be used and the minimization of risks” (E6).*
Discuss with the family the care to be developed in partnership	*“If the family is willing, if the patient, being aware and oriented, also agrees, it is a very advantageous partnership for everyone.” (E3)* *“I, for example, always ask how they want to do with the bath because many of the patients even prefer to take it at the end of the day with the family.” (E3).*
Prepare discharge with the family from the beginning of hospitalization	*“… when I go to position … it is timely to empower that family, I can see what the difficulties are and I’m showing that it is possible. But if I draw the curtain, I’m creating a mystery, I’m increasing the difficulty and I’m not creating a relationship of trust … Nurses always have a lot to do, but when the daughter is there, I can take advantage of the moment of care provision and include the family … I’ve also always encouraged mobility, whether in a wheelchair, a walk to the hospital garden or walking. And in this, families feel useful and are helping. There are things that I wish I could do, but with the family helping me, they can do little things that I wish I could do, but I can’t. And in these little things we will prepare for the baby. And with these little things we are already preparing for discharge” (E10).* *“There is an issue that really affects me a lot which is the moment of discharge, I think that it is ungrateful that we send the relatives home in a different condition from the one they were in before coming to the hospital and I, the nurse, don’t have any time to talk to and be with that family and explain and try to understand what the person feels. Basically, we are almost pushing a problem into their hands and expecting them to manage because now we don’t have time to talk to them and this, as a team, we have been trying to work towards this for some time because the medical team sometimes doesn’t even inform us of the patients’ discharge and we know from the family member who comes to get them. So we can’t work or empower the family members and we know that many of them would even appreciate that contact and we don’t have the opportunity. ”(E11). * *“in stroke patients, when he presents gait alterations, I alert the family to the need for changes that may be necessary at home, perhaps due to my training in rehabilitation. Or, when I go to give medication, I take the opportunity to ask how the room is, I try to assess the need to reorganise the space, such as removing some rugs so that the patient doesn’t fall. On this topic I talk about reorganising the spaces, greater amplitudes, unilateral gait support, these safety issues” (E3).*
Teaching/Informing the family	*“…we, as health team, have the role of teaching and promoting the family as a safety promoter. I think that the presence of families has more advantages than disadvantages and we should be elements promoters of their presence in order to provide the safety of hospitalized patients. We should teach them so that their conduct does not compromise safety in the provision of care”. (E10) * *“If we raise awareness, if we teach, there is potential in families, there are only positive aspects in having families in hospitals with their ill family members. ” (E3)* *“For me, the presence of the family or caregiver is always a formative moment” (E5).* *“Training could be done with groups of families and it could be only 5 or 10 min to give information on hospital behaviour… This should be included in one of the routines, for example in the afternoon shift, it makes perfect sense to raise awareness about the hospital environment. And then we’ll save a lot of questions afterwards…It’s very much our area…” (E3).*
Providing family participation in care	*“… when it’s a child, everyone really encourages mothers and fathers to participate in care. I think we should do a bit of that in all the services with our service users because then they go home. I remember having collaborated with a mother of an adult patient with special needs in hygiene care and she was the one who guided me in her son’s preferences or needs. With the elderly you don’t see this. We should encourage more family as participants in care”. (E3).*
Supervising/following up the family in hospital by nurses	*“…it should be the nurse who accompanies the visits, we are the most knowledgeable professional for this” (E7)* *“We always try to receive relatives at the entrance to the unit and then reinforce the instructions on hand hygiene, avoid touching hospital objects and equipment, try to give some instructions, particularly about what he/she was at home: the bags, clothes, give some advice and then I accompany the relative to the patient, validate again some information on hand hygiene, contact with the patient, so that he/she can also understand that if the relative is not so close or does not touch, it is for guidance” (E7).*
Encouraging the family to report safety issues	*“I encourage family members and the patients themselves to report any unsafe event or circumstance. It’s fundamental, it’s that story, both the family member and the patient are the first source of verification. … it is up to the nurse to promote that the person expresses all his doubts… I encourage the family in this sense. When the family asks “My relative isn’t taking medication now?”, then I sit down to verify and explain to the family the compliance with the therapeutic scheme. And sometimes the patients and/or families are right. They are a source of security for our practice… It’s like the situation of a family member saying ‘my mother’s name is Maria Alice and she has a bracelet with another name’, it happened to me a few years ago. ”(E10)*
Make video calls	*“It is a hospital project that we can make video calls with the patient’s family or significant others, whenever requested by the patient or the family members, so, whenever possible, on a daily basis and whenever necessary, we make video calls to try to bring the patient closer to the family” (EI5).* *“getting a mobile phone with video call, normally our elderly patients are not used to this technology, to be able to see the family member” (E9)*
Develop leadership influencing PFCC	*“There is a lack of reflection, I am very sorry that our bosses don’t contribute to that. Because the born leader has to con-vide to reflect. I sometimes say this nurse needs to be worked on. Help him/her to be a better nurse, better in care and with the family…” (E10)*
Promote safety culture in the hospital	*“Behaviour generates behaviour. So, who empowers people is us. If I have a proper behaviour, the family will have a proper behaviour. It’s much easier to say that it’s someone else’s problem. But what do I do to model the behaviour of others? The source of noise in hospitals is not the non-professional families. It is much easier to say that the problem is the families” (E10).*
Plan training for nurses on family involvement/communication attitude	*“I think that what is essential is a very strong investment in the professionals’ communication skills. I dare say that only about 25% of nurses have an attitude of involvement with the family, of availability. I think hospitals should invest a great deal in improving the communication skills of professionals. My big question is what type of training because the classic training brings us nothing” (E10).*
Improve nursing staffing	*“We have to be realistic, nursing allocations in medical inpatient services are not always the most adequate, so as much as we want to increase the contact time with families, we don’t always succeed. And I’m saying this because I really value support and the family as a partner.” (E2)*
Create spaces outside the ward for visits	*“It was important to create specific spaces to receive visits, to have their own place where people could go if the patient could not go to that place, then we would have to manage it in another way. But whenever possible, when the client left the ward, he could receive his visit in another place” (E7).*
Stimulate the primary nursing method	*“… in the service, we use the work method of the reference nurse, and the work method itself ends up being a facilitating strategy for approaching the family. I think that the method ended up being favourable in this phase, it supported the family and brought them closer. They also felt more at ease to express their concerns, because they know that I, being the reference nurse, feel free to telephone and say “I’d like to speak to nurse x”, and there is a greater continuity after the information that is provided on the health condition of the patient” (E5).*
Stimulate family care centered on the family member of reference	*“Yes, we encourage families to appoint a spokesperson, even the transmission of information is usually transmitted almost exclusively to the family member signalled by the patient… We always communicate with the family member who was initially signalled. So we try to channel the information” (E7).*
Provide lockers for the family	*“An important strategy was a place for the family to keep their personal belongings. We have facilities that sometimes, even for teeth, are not the best facilities, but it was important to have a locker place where families could leave their coats, their wallets and come with as few things as possible from outside to the patient’s bedside. What happens is that the wallets that are in the street and on the ground are then found, for example, on the patients’ beds. I think that this was undoubtedly important” (E1).*
Create checklist for teaching the family	*“There could be a teaching protocol, a checklist for this situation.” (E4)*
Create a guide for the family in the hospital	*“I think the hospital can create rules of conduct that besides stating what should be done and cannot be done in the hospital. It should also provide a script of how to be family in the hospital. A script of good practices and good ways of being family in the hospital.” (E2).*
Visits management	*“The decision on visits is usually a medical decision and in terminal situations or other more delicate situations …, that’s what my service is doing.” (E3) “The nurse is the health care professional who is in the inpatient unit 24 h a day and, therefore, should be responsible for managing visits” (E2)* *The decision on visits is usually a medical decision and in terminal situations or other more delicate situations …, that’s what my service is doing.” (E3)* *“It makes no difference to me whether it is the doctor or the nurse who authorises the visits, it must be with valid justification. I think that ideally it should be the doctor and nurse as a team.” (E4)*
Separate visiting hours for family member of reference	*“the main caregiver or the direct relative who could have more time than the remaining visits.” (E8)*
Extended visiting hours	*“I think it would be really interesting to have a longer schedule because not everyone has the same opportunities to manage their time according to the visiting schedule” (E7)*
24 h visiting policy	*“There are hospitals in other countries that allow 24 h visits, there are open-door hospitals, this seems surprising to us, but culturally, for them, it is unacceptable that relatives are in the hospital without their families and we say that we are a very family-oriented people and, in the end, we keep families away a little bit.” (E10).*

## Data Availability

The data presented in this study are available on request from the corresponding author due to privacy, and ethical reasons.

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
