# Peer review of "Patient and Family-Centered Care to Promote Inpatient Safety: An Exploration of Nursing Care and Management Processes"

_nursrep, 2025, doi:10.3390/nursrep15070260_

Round 1
Reviewer 1 Report
Comments and Suggestions for Authors
- The introduction should more clearly articulate the theoretical linkage between PFCC and patient safety, and differentiate the study from previous literature.
- The nursing process stages of diagnosis and evaluation are notably absent from the findings and discussion. The authors should clarify why these stages were not represented and interpret their absence within the context of PFCC implementation.
- Some of the strategies presented (providing family lockers, creating designated visiting areas) lack strong empirical support. These should be explicitly identified as emerging or exploratory practices that warrant further validation.
- The distinction between the care process and logistic process is at times unclear. Several strategies seem to overlap conceptually between these two categories. Clearer definitions and criteria for categorization are recommended.
- The results and discussion sections contain multiple repetitive statements and quotations, which reduce the readability of the manuscript. Consolidating and streamlining the narrative would enhance clarity and flow.
1. The manuscript contains overly long and occasionally repetitive sentences, especially within the Results and Discussion sections. This diminishes overall readability and weakens the coherence of the argument.
2. Additionally, certain phrases and grammatical structures lack the level of academic precision expected in scholarly writing and would benefit from careful revision.
3. The use of transitional phrases and logical connectors is at times inadequate or inconsistent, which disrupts the narrative flow and may hinder reader comprehension.
Author Response
We would like to thank the reviewer for his comments and suggestions, which we consider to be extremely pertinent and valuable for improving our manuscript. Below are the detailed responses to each of the comments, as well as the changes made to the article.
Comments 1 - The introduction should more clearly articulate the theoretical linkage between PFCC and patient safety, and differentiate the study from previous literature.
Response 1 - Thank you very much for your suggestion. We've made some changes to the introduction, for example: “The Global Patient Safety Action Plan 2021–2030 recognizes that the involvement of patients and families in the care process is essential to achieving safer care. It offers concrete recommendations, including involving patients and families in the formulation of patient safety policies and ensuring the transparent sharing of information. (World Health Organization, 2021) A meta-analysis of 22 studies on patient and family engagement interventions demonstrated a significant reduction in adverse events, a decrease in length of hospital stay, improvements in patients’ safety experience, and increased satisfaction with care.(Giap & Park, 2021)”…” The available evidence evaluates patient and family involvement only at the level of direct health care delivery.(Cai et al., 2024) However, there are few published studies on the involvement of patients and families in patient safety (Abdi et al., 2024). The application of the PFCC model in adult care requires further research and implementation when compared to its use in pediatric settings. Given that nurses are the health professionals who maintain the closest and most continuous contact with families, their behaviors and attitudes can significantly influence the hospitalization experience of both patients and their families, as well as key outcomes related to care quality and safety. Recent evidence underscores that nurses, as the primary formal caregivers, should play a leading role in promoting patient and family involvement in safety-related practices(Cai et al., 2024).
Research on patient and family involvement in patient safety remains limited, particularly at the organizational and system levels.(Cai et al., 2024; Seyfulayeva et al., 2025) Therefore, the aim of this study is to understand the process of family care in the hospital setting as developed by nurses, and to explore the logistical processes that support the implementation of family-centered care to ensure patient safety during hospitalization.
Comments 2- The nursing process stages of diagnosis and evaluation are notably absent from the findings and discussion. The authors should clarify why these stages were not represented and interpret their absence within the context of PFCC implementation.
Response 2 -
Comments 3- Some of the strategies presented (providing family lockers, creating designated visiting areas) lack strong empirical support. These should be explicitly identified as emerging or exploratory practices that warrant further validation.
Response 3 – Thank you very much for your suggestion. We've made some changes to the discussion Lines 355-359
Comments 4 - The distinction between the care process and logistic process is at times unclear. Several strategies seem to overlap conceptually between these two categories. Clearer definitions and criteria for categorization are recommended.
Response 4 – Thank you very much for your suggestion. We've made some changes :”The analysis of the interview data revealed two main thematic families: the Care Process and the Logistic Process, which emerged as the two major strategic domains for implementing Patient- and Family-Centred Care (PFCC) with a focus on patient safety (Table 2). The Care Process encompasses a series of interventions related to patient diagnosis and treatment, aimed at providing optimal care. (Seys et al., 2013) Studies have shown that improving the organisation of care delivery contributes to enhanced patient safety in healthcare settings. (Seys et al., 2013) Furthermore, there is a strong correlation between the quality of the care process and the centrality of the patient and family in care. (Seys et al., 2013; Vanhaecht et al., 2007) In contrast, the Logistic Process in healthcare refers to the coordination of complex operations involving multiple people, facilities, and resources within the health system. (Božić et al., 2022) It also plays a crucial role in supporting the quality of the care process. (Božić et al., 2022)”
Comments 5 - The results and discussion sections contain multiple repetitive statements and quotations, which reduce the readability of the manuscript. Consolidating and streamlining the narrative would enhance clarity and flow.
Response 5 –
Comments on the Quality of English Language:
- The manuscript contains overly long and occasionally repetitive sentences, especially within the Results and Discussion sections. This diminishes overall readability and weakens the coherence of the argument.
- Additionally, certain phrases and grammatical structures lack the level of academic precision expected in scholarly writing and would benefit from careful revision.
- The use of transitional phrases and logical connectors is at times inadequate or inconsistent, which disrupts the narrative flow and may hinder reader comprehension.
Response: Thanks for the suggestion for improvement, the whole article has been revised by an English teacher and it really is much better.
Reviewer 2 Report
Comments and Suggestions for Authors
- In the introduction, authors should include more data or statistics to highlight the difference between what is currently done and the recommended PFCC practices, and they should add more recent references (especially from 2022–2024) from around the world to support the reasons for the study.
- Methods:
- Clarify the sampling criteria and demographic profile of the participants (e.g., years of experience, unit type).
- Provide the interview guide or sample questions in an appendix or supplementary file.
- Explain how data saturation was assessed.
- Provide ethical considerations more explicitly (e.g., IRB approval, informed consent).
- Results: Authors need tables or figures to summarize categories and sample quotes to strengthen the presentation and enhance clarity.
- Conclusion
- More clearly relate findings to existing literature.
- Suggest practical implications for policy or training based on the topics.
- Acknowledge limitations of the study in more detail (e.g. transferability due to small sample size).
Author Response
We would like to thank the reviewer for his comments and suggestions, which we consider to be extremely pertinent and valuable for improving our manuscript. Below are the detailed responses to each of the comments, as well as the changes made to the article.
Comments 1 - In the introduction, authors should include more data or statistics to highlight the difference between what is currently done and the recommended PFCC practices, and they should add more recent references (especially from 2022–2024) from around the world to support the reasons for the study.
Response 1 – Thank you for suggesting this improvement. We've made some changes to the introduction, for example: “The Global Patient Safety Action Plan 2021–2030 recognizes that the involvement of patients and families in the care process is essential to achieving safer care. It offers concrete recommendations, including involving patients and families in the formulation of patient safety policies and ensuring the transparent sharing of information. (World Health Organization, 2021) A meta-analysis of 22 studies on patient and family engagement interventions demonstrated a significant reduction in adverse events, a decrease in length of hospital stay, improvements in patients’ safety experience, and increased satisfaction with care.(Giap & Park, 2021)”
” The available evidence evaluates patient and family involvement only at the level of direct health care delivery.(Cai et al., 2024) However, there are few published studies on the involvement of patients and families in patient safety (Abdi et al., 2024). The application of the PFCC model in adult care requires further research and implementation when compared to its use in pediatric settings. Given that nurses are the health professionals who maintain the closest and most continuous contact with families, their behaviors and attitudes can significantly influence the hospitalization experience of both patients and their families, as well as key outcomes related to care quality and safety. Recent evidence underscores that nurses, as the primary formal caregivers, should play a leading role in promoting patient and family involvement in safety-related practices(Cai et al., 2024).
Research on patient and family involvement in patient safety remains limited, particularly at the organizational and system levels.(Cai et al., 2024; Seyfulayeva et al., 2025) Therefore, the aim of this study is to understand the process of family care in the hospital setting as developed by nurses, and to explore the logistical processes that support the implementation of family-centered care to ensure patient safety during hospitalization..”
Comments 2 - Methods:- Clarify the sampling criteria and demographic profile of the participants (e.g., years of experience, unit type).
Response 2 – Thank you for suggesting this improvement. This information is in the methods section as shown below:
“Participants were nurses working in internal medicine and surgical wards across four hospitals in northern Portugal. They were selected through convenience sampling, as they were the most accessible and met the pre-established inclusion criteria. These criteria included: being in active clinical practice at the time of the interview; having experience in internal medicine and/or surgery; having at least four years of hospital-based professional experience; and being available to participate.”
Comments 3 - Provide the interview guide or sample questions in an appendix or supplementary file.
Response 3 – Thanks for the suggestion. We've moved this information to Appendix A of the article.
Comments 4 - Explain how data saturation was assessed.
Response 4 – Thank you for suggesting this improvement. It is described in the methods as follows: “After the seventh interview, no new data emerged that contributed to the development of additional categories. Therefore, after completing ten interviews, data saturation was considered to have been reached. (Hennink & Kaiser, 2022; Saunders et al., 2018)”
Comments 5 - Provide ethical considerations more explicitly (e.g., IRB approval, informed consent).
Response 5 – Thank you for suggesting this improvement. This information is in the methods section as shown below:
“All ethical and legal principles governing research involving human participants were rigorously respected throughout the study. Ethical approval was obtained from the Joint Ethics Committee of the Porto Hospital and University Centre and the Abel Salazar Biomedical Sciences Institute (ICBAS) at the University of Porto (UP) – (protocol code 2020/CE/P010(P320/CETI/ICBAS. The study procedures strictly upheld the principles of participant anonymity, confidentiality, and voluntary informed consent. Furthermore, the research was conducted in full compliance with the ethical standards outlined in the Declaration of Helsinki. Participants received detailed information about the study’s objectives, procedures, potential risks and benefits, confidentiality measures, and their right to withdraw at any time without penalty.”
Comments 6 - Results: Authors need tables or figures to summarize categories and sample quotes to strengthen the presentation and enhance clarity.
Response 6 – Thank you very much for your suggestion. We've made some changes to the results (pages 5-8)
Comments 7 – Conclusion: More clearly relate findings to existing literature.
Response 7 – Thank you very much for your suggestion. We've made some changes to the conclusion (lines 534-540)
Comments 8 - Suggest practical implications for policy or training based on the topics.
Response 8 – Thank you very much for your suggestion. We've made some changes to the conclusion with implications for policy or training based on the topics (Lines 519-524)
Comments 9 - Acknowledge limitations of the study in more detail (e.g. transferability due to small sample size).
Response 9 – Thank you very much for your suggestion. We've made some changes to the conclusion (Lines 525-628)
Reviewer 3 Report
Comments and Suggestions for Authors
Thank you for the opportunity for reviewing this paper on an important subject. Although the research method seems sound and the results could be very informative, the reporting needs substantial improvement.
Introduction:
- P1L39 >>> It is now somewhat unclear what you mean by “The latter …”, please specify that this refers to the family.
- P1L41-42 >>> “This contribution … own health”, this statement needs a reference.
Theoretical Framework:
- P2L83-85 >>> You begin this statement with “Published studies …” but cite only one, which seems to be this groups earlier work. Please give a more comprehensive overview on the current knowledge on this matter.
- P3L101-102 >>> The second part of the objective (… explore the logistic processes identified …) is not supported by the information in the theoretical framework section. Please elaborate more on that subject.
Materials and Methods:
- Please use the COREQ-statement to structure your manuscript. You state that you do on P3L145-146, but an awful lot of information is missing.
- P3L116-117 >>> Please elaborate on what you mean by “9 nurses and 1 nurse”. Wouldn’t that just make 10 nurses?
- Please add information on the reflexivity, the research team and who performed the specific tasks around gathering and analysing data.
- The italic information between brackets (codes, categories, families) does not add to the comprehension of the paper. So pick a framework, adopt its language and stick with that.
- Please add the code tree as an image to guide the reader trough your findings.
Results:
- The Results-section is too brief to understand the process. Please elaborate more on the results and add quotes and other sources of information to present rich data.
Discussion:
- The Discussion contains a lot of information, but the structure is poor. This leads to poor readability and a I reader, I get lost along the way about the point the authors try to make. Please restructure into clear sections, put the findings in the results-section, add a framework drawing and highlight quotes between sections. In the current form, the Conclusions cannot the drawn from the text in the Discusson.
Author Response
We would like to thank the reviewer for his comments and suggestions, which we consider to be extremely pertinent and valuable for improving our manuscript. Below are the detailed responses to each of the comments, as well as the changes made to the article.
Comments 1 -P1L39 >>> It is now somewhat unclear what you mean by “The latter …”, please specify that this refers to the family.
Response 1 – Thank you very much for pointing out this need for correction. We have corrected it as follows: “The family can play a key role in contributing to the patient’s emotional stability and providing support across various domains”
Comments 2 - P1L41-42 >>> “This contribution … own health”, this statement needs a reference.
Response 2 – Thank you very much for pointing out this need for correction. We have corrected it as follows: “This involvement also has a positive impact on the family's own well-being, as it fulfils their need to support the patient, access information directly, and be present with their hospitalized relative”
Comments 3 - Theoretical Framework: P2L83-85 >>> You begin this statement with “Published studies …” but cite only one, which seems to be this groups earlier work. Please give a more comprehensive overview on the current knowledge on this matter.
Response 3 – Thank you very much for pointing out this need for correction. We have corrected it as follows: “Published studies also show that families often act to protect the safety of their hospitalized relatives and, in some cases, succeed in following patient safety recommendations. (T. Correia et al., 2023; T. S. P. Correia et al., 2020; Oyesanya & Bowers, 2017) However, families frequently report feeling unprepared and lacking guidance and collaboration from health professionals(T. S. P. Correia et al., 2022a; T. S. P. Correia et al., 2020).”
Comments 4 - P3L101-102 >>> The second part of the objective (… explore the logistic processes identified …) is not supported by the information in the theoretical framework section. Please elaborate more on that subject.
Response 4 – – Thank you very much for pointing out this need for correction. We have corrected it as follows:” The available evidence evaluates patient and family involvement only at the level of direct health care delivery.(Cai et al., 2024) However, there are few published studies on the involvement of patients and families in patient safety (Abdi et al., 2024). The application of the PFCC model in adult care requires further research and implementation when compared to its use in pediatric settings. Given that nurses are the health professionals who maintain the closest and most continuous contact with families, their behaviors and attitudes can significantly influence the hospitalization experience of both patients and their families, as well as key outcomes related to care quality and safety. Recent evidence underscores that nurses, as the primary formal caregivers, should play a leading role in promoting patient and family involvement in safety-related practices(Cai et al., 2024). Research on patient and family involvement in patient safety remains limited, particularly at the organizational and system levels.(Cai et al., 2024; Seyfulayeva et al., 2025) Therefore, the aim of this study is to understand the process of family care in the hospital setting as developed by nurses, and to explore the logistical processes that support the implementation of family-centered care to ensure patient safety during hospitalization.”
Comments 5 - Materials and Methods: Please use the COREQ-statement to structure your manuscript. You state that you do on P3L145-146, but an awful lot of information is missing.
Response 5 – thank you very much for pointing out this suggestion for improvement. Several revisions were made throughout the methodology section to highlight this information.
Comments 6 - P3L116-117 >>> Please elaborate on what you mean by “9 nurses and 1 nurse”. Wouldn’t that just make 10 nurses?
Response 6 – Thank you very much for pointing out this need for correction. We have corrected it as follows: “Based on the criterion of theoretical saturation, a total of 10 participants were included: 9 female nurses and 1 male nurse, aged between 28 and 62 years (mean age 39 years).”
Comments 7 - Please add information on the reflexivity, the research team and who performed the specific tasks around gathering and analysing data.
Response 7 – The data analysis was carried out by T.C. and M.M., and the reflection on the findings was further expanded through discussions with F.B. and O.V.
Comments 8 - The italic information between brackets (codes, categories, families) does not add to the comprehension of the paper. So pick a framework, adopt its language and stick with that.
Response 8 – Thank you very much for pointing out this need for correction. I tis done.
Comments 9 - Please add the code tree as an image to guide the reader trough your findings.
Response 9 – Thank you very much for pointing out this need for correction. I tis done.
Comments10 - Results: The Results-section is too brief to understand the process. Please elaborate more on the results and add quotes and other sources of information to present rich data.
Response 10 – Thank you very much for pointing out this need for correction. I tis done.
Comments 11 - Discussion: The Discussion contains a lot of information, but the structure is poor. This leads to poor readability and a I reader, I get lost along the way about the point the authors try to make. Please restructure into clear sections, put the findings in the results-section, add a framework drawing and highlight quotes between sections. In the current form, the Conclusions cannot the drawn from the text in the Discusson.
Response 11-Thank you very much for pointing out this need for correction. I tis done. The whole discussion has been reformulated
Thanks for the suggestion, the whole article has been revised by an English teacher and it really is much better.